# The Influence of Obesity on Outcomes with Immune Checkpoint Blockade: Clinical Evidence and Potential Biological Mechanisms

**DOI:** 10.3390/cells12212551

**Published:** 2023-10-31

**Authors:** Andrew W. Hahn, Neha Venkatesh, Pavlos Msaouel, Jennifer L. McQuade

**Affiliations:** 1Department of Genitourinary Medical Oncology, Division of Cancer Medicine, The University of Texas MD Anderson Cancer Center, Houston, TX 77030, USA; pmsaouel@mdanderson.org; 2Department of Internal Medicine, Baylor College of Medicine, Houston, TX 77030, USA; 3Department of Translational Molecular Pathology, The University of Texas MD Anderson Cancer Center, Houston, TX 77030, USA; 4David H. Koch Center for Applied Research of Genitourinary Cancers, The University of Texas MD Anderson Cancer Center, Houston, TX 77030, USA; 5Department of Melanoma Medical Oncology, Division of Cancer Medicine, The University of Texas MD Anderson Cancer Center, Houston, TX 77030, USA

**Keywords:** obesity, body composition, immunotherapy, melanoma, renal cell carcinoma

## Abstract

Immune checkpoint blockade (ICB) is a mainstay of treatment for advanced cancer, yet tumor response and host toxicity are heterogenous in those patients who receive ICB. There is growing interest in understanding how host factors interact with tumor intrinsic properties and the tumor microenvironment to influence the therapeutic index with ICB. Obesity, defined by body mass index, is a host factor associated with improved outcomes in select cancers when treated with ICB. While the biological mechanism for this obesity paradox is not fully understood, pre-clinical and translational studies suggest obesity may potentially impact tumor metabolism, inflammation, and angiogenesis. Herein, we summarize clinical studies that support an obesity paradox with ICB, explore potential biological mechanisms that may account for the obesity paradox, and address methodological challenges to consider when studying obesity and treatment outcomes.

## 1. Introduction

Immune checkpoint blockade (ICB) refers to treatments, typically monoclonal antibodies, that target immune checkpoints expressed on the surface of immune cells to increase immune recognition and clearance of tumor cells [1]. The immune checkpoints targeted may be inhibitory checkpoint molecules, such as cytotoxic T lymphocyte-associated protein 4 (CTLA-4), programmed death 1 (PD-1), and lymphocyte activation gene-3 (LAG-3), or co-stimulatory checkpoint molecules, such as ICOS, OX40, and 4-1BB [2]. Historically, advanced cancer was primarily treated with cytotoxic chemotherapy, and in the early 2000s, therapies that target oncogenic signaling pathways were added [3]. Ipilimumab, a monoclonal antibody targeting CTLA-4, was approved for the treatment of advanced melanoma in 2011 [4]. This marked the beginning of the ICB era, and ICB is now a mainstay for treatment of advanced cancer with more than 50 FDA approvals across cancer types [4]. ICB is notable among cancer therapies for the durability of the clinical responses it produces due to immune memory. However, ICB can also induce serious toxicities due to activation of the immune system against normal tissues.

Outcomes with ICB are heterogeneous, in terms of both tumor response and toxicity. Accordingly, there is a large body of research investigating determinants of efficacy and toxicity with ICB across three compartments: tumor intrinsic properties, the tumor microenvironment, and the host ecosystem. Tumor intrinsic factors that can influence the efficacy of ICB include genomic and epigenetic alterations to pathways critical to the immune response, such as antigen presentation via MHC-1, IFN-γ signaling, or neoantigen production [2,5,6,7,8]. The tumor microenvironment (TME) comprising immune cells, stromal cells, and vasculature has also been shown to influence the efficacy of ICB. For example, cancer-associated fibroblasts can exclude T cells from the TME leading to resistance, and myeloid cells such as tumor-associated macrophages and myeloid-derived suppressor cells can contribute to an immunosuppressive TME [9,10,11]. The third component influencing the efficacy of ICB is the host ecosystem that likely interacts with tumor intrinsic properties and the TME, and can shape the cancer-immune setpoint [12]. These host factors include both unmodifiable factors (age and biological sex), as well as potentially modifiable factors (obesity, diet, exercise) and the human microbiota that are shaped by these factors [13,14,15,16,17].

Obesity, defined as a body mass index (BMI) of 30 or higher, is associated with an increased risk of developing 13 different cancers in addition to its association with diabetes mellitus, cardiovascular disease, and death [18]. However, in terms of outcomes in patients who have already been diagnosed with cancer, numerous studies have reported improved outcomes for obese patients receiving ICB and targeted therapy when compared to patients with a normal BMI. The first major study to report this finding with ICB was led by our group, and in 538 patients with metastatic melanoma who received ICB, obesity was associated with a 25% reduction of the hazard for disease progression and a 36% reduction of the hazard for death [17]. Multiple studies have confirmed this finding and led to use of the term “obesity paradox” where obesity is paradoxically associated with improved outcomes with ICB. The phenomenon of an obesity paradox is not unique to patients with cancer who receive ICB as similar findings have been described with cardiometabolic disease [19,20]. Whether the obesity paradox points to differential biology or results from methodological issues remains an area of active controversy [21].

In this narrative review, we will first examine the clinical studies that support an obesity paradox in ICB-treated cancer patients, including pan-cancer analyses as well as the three cancer types in which this question has been studied in depth: melanoma, renal cell carcinoma (RCC), and non-small cell lung cancer (NSCLC). Then, we will explore potential biological mechanisms that may account for the obesity paradox and address methodological challenges to consider when studying obesity and treatment outcomes. Finally, we will discuss potential future studies to elucidate the cause of the obesity paradox and the influence of the host ecosystem on clinical outcomes for advanced cancers.

## 2. Clinical Evidence for an Obesity Paradox with ICB

Multiple clinical studies demonstrate that obese or overweight patients with advanced or metastatic melanoma, RCC, and NSCLC have improved survival when treated with ICB, so we summarized the clinical evidence in support of an obesity paradox with ICB for these malignancies.

### 2.1. Melanoma

Our group initially identified a positive association between obesity and survival outcomes in two cohorts of patients with metastatic melanoma treated with ICB, including a randomized controlled trial of ipilimumab plus dacarbazine [n = 207] and a retrospective cohort treated with pembrolizumab, nivolumab, or atezolizumab [n = 331] [17]. Across these cohorts, patients with a BMI ≥ 30 had improved progression-free survival (PFS, HR 0.75, 95% CI 0.56–1.00) and overall survival (OS, HR 0.64, 95% CI 0.47–0.86, Table 1). Notably, we found a difference in the association of BMI and ICB outcome when stratified by sex. Obese males receiving ICB had a significantly lower risk of disease progression (PFS HR 0.62, 95% CI 0.41–0.92) and death (HR 0.62, 95% CI 0.39–0.98, Table 1), but this signal was attenuated for females treated with ICB. A retrospective study of 139 patients with advanced melanoma evaluated BMI classes and serum creatinine, as a surrogate for muscle mass, to begin investigating how body composition phenotypes may influence the obesity paradox with ICB [22]. The most pronounced benefit was observed in the overweight or class I obese cohort for PFS (HR 0.43, 95% CI 0.19–0.95) and OS (HR 0.26, 95% CI 0.10–0.71, Table 1). In complex exploratory modeling using BMI, creatinine, and clinical variables, the longest survival was seen in overweight or class I obese patients with a serum creatinine > 0.9. The authors suggest that having increased skeletal muscle mass may confer a survival advantage and contribute to the obesity paradox. Another retrospective analysis of 76 patients from Austria with metastatic melanoma who received the CTLA-4 monoclonal antibody ipilimumab reported a higher objective response rate in the overweight group (n = 46, BMI ≥ 25) and a weak signal favoring longer OS (HR 1.81, 95% CI 0.98–3.33) in the overweight group compared to the normal BMI group (n = 40, BMI < 25) [23]. However, the PFS association between overweight and normal BMI patients was inconclusive (HR 1.03, 95% CI 0.62–1.70). Additionally, a large, retrospective analysis of 446 patients with metastatic melanoma who received ICB found no impact of BMI on OS (HR 1.02, 95% CI 0.99–1.05, Table 1) nor PFS (HR 1.00, 95% CI 0.98–1.03) [24].

Given the limitations of BMI as a surrogate of body composition, a retrospective study of 84 patients who received ipilimumab evaluated the association between body composition measurements on computed tomography (CT) and clinical outcomes with ICB [25]. Baseline sarcopenia and low muscle quality were not associated with shorter survival, but loss in muscle mass during the course of treatment adversely affected survival outcomes. The group of patients with highest muscle loss of ≥7.5%/100 days had significantly lower OS compared to those with muscle loss <7.5%/100 days (HR 2.14, 95% CI 1.06–4.28). A larger, retrospective analysis evaluated the impact of body composition measures on 287 patients treated with ICB [26]. The results were inconclusive regarding the association of BMI with clinical response, and the only body composition parameter significantly associated with PFS in univariable analyses was sarcopenic obesity (HR 1.47, 95% CI 1.02 to 2.12, *p* = 0.037).

### 2.2. Renal Cell Carcinoma

ICB combinations with or without targeted therapy are currently the standard first-line treatment for metastatic clear cell RCC. In a retrospective cohort of 203 patients with metastatic clear cell RCC who received ICB, obesity was associated with improved OS when compared to normal BMI patients in unadjusted analyses (HR 0.54, 95% CI 0.31–0.95, Table 1), but the association was lost when controlling for an RCC prognostic risk score (HR 0.72, 95% CI 0.40–1.30) [27]. In a retrospective study of 735 patients with metastatic RCC who received ICB from the International Metastatic RCC Database Consortium (IMDC), patients with overweight or obese BMI had favorable OS compared with those with normal or low BMI (HR 0.75, 95% CI 0.57–0.97, Table 1) [28]. However, time to treatment failure (TTF) and ORR were not significantly improved in those patients with overweight and obese BMI (TTF HR 0.98, 95% CI 0.80–1.20). The retrospective, multicenter ARON-1 study included 675 patients with metastatic RCC treated with first-line ICB combinations with or without targeted therapy, and they found that overweight and obese patients (BMI > 25) had significantly higher OS (55.7 vs. 28.4 months, *p* < 0.001, Table 1) [29]. In this cohort, overweight or obese BMI was not associated with a significant improvement in PFS (15.9 vs. 14.1 months, *p* = 0.07). These studies of BMI and ICB in metastatic RCC do not show a significant association between obesity and more direct measures of response to therapy (PFS, TTF, ORR).

A retrospective study evaluated the influence of radiographic measures of body composition on clinical outcomes in 205 patients with metastatic clear cell RCC who received ICB [30]. High skeletal muscle mass index (SMI) was associated with improved OS (low vs. high SMI HR 1.65, 95% CI 1.13–2.43), but adiposity measures were not associated with OS. Body composition variables also were not associated with ORR or PFS. In contrast, body composition parameters were not associated with OS in a retrospective study of 99 patients receiving first-line nivolumab plus ipilimumab, and low SMI and subcutaneous adiposity were associated with improved PFS [31]. Finally, a meta-analysis was conducted to investigate whether the obesity paradox holds true for patients with metastatic RCC who received ICB (n = 2281) [32]. They found that overweight and obese BMI was associated with improved OS (HR 0.77, 95% CI 0.65–0.91) and PFS (HR 0.66, 95% CI 0.44–1.00). The authors suggest that improved nutritional status and chronic inflammation in the obese state may account for this obesity paradox.

### 2.3. Non-Small Cell Lung Cancer

Obese patients with NSCLC have shown a similar positive response to ICB, especially those treated with monoclonal antibodies targeting the PD-1/PD-L1 axis. In two studies, the degree of PD-L1 expression influenced the relationship between BMI and survival. An international retrospective study found an increased objective response rate (ORR, OR 1.61, 95% CI 1.04–2.50) as well as longer PFS (HR 0.61, 95% CI 0.45–0.82) and OS (HR 0.70, 95% CI 0.49–0.99, Table 1) in baseline obese patients with metastatic NSCLC compared to normal BMI patients who had PD-L1 expression ≥50% and were treated with first-line pembrolizumab (n = 962) [33]. The authors speculated this may be due to obesity conferring greater functional reserve or obesity leading to higher doses of ICB when weight-based dosing is employed. Similarly, a pooled analysis of four clinical trials found a nearly linear, positive association between BMI and OS in patients with advanced NSCLC who received PD-L1 inhibitor atezolizumab (n = 1434) [34]. The cohort with the highest PD-L1 expression who also had obesity (BMI ≥ 30) had the lowest hazard for death (HR 0.48, 95% CI 0.34–0.66) and for progression (HR 0.78, 95% CI 0.62–0.96, Table 1). A study suggested that the line of therapy when a PD-1/PD-L1 inhibitor is administered influences the survival advantage of obesity. In 513 patients with advanced NSCLC, longer survival was observed only in high BMI patients (BMI > 22 per regional ideal BMI) treated with second- or later-line PD-1/PD-L1 inhibitors [35]. However, the survival signal between high and low BMI patients who received first-line pembrolizumab therapy was inconclusive, which raises the question of whether BMI was a surrogate for cancer cachexia in this cohort. Again, the association between PD-L1 expression, BMI, and response to ICB was observed, as median PFS in patients with high BMI and PD-L1  ≥ 50% was 17.0 months but only 9.4 months for all patients with PD-L1  ≥ 50%.

A theme we see in these studies is the importance of evaluating anthropometric measures of body mass, instead of BMI alone. In a study of 55 patients who received nivolumab for metastatic NSCLC, pretreatment 18F-FDG PET/CT scans were used to objectively measure body composition parameters [36]. In a multivariable analysis incorporating clinical parameters, patients with low subcutaneous fat mass had inferior survival when treated with nivolumab (HR 0.75, *p* = 0.006). This suggests that the subcutaneous adipose tissue may be contributing to the obesity paradox observed in advanced NSCLC.

**Table 1 cells-12-02551-t001:** Title: Summary of clinical studies evaluating the influence of obesity by body mass index (BMI) on clinical outcomes with immune checkpoint blockade (ICB). Legend: N = total number, PFS = progression-free survival, * = Or other similar time to next treatment endpoints if PFS unavailable, CI = confidence interval, OS = overall survival, NA = not available.

Author	BMI Categories	N	PFS HR * (95% CI)	OS HR (95% CI)
Melanoma				
McQuade JL, et al. [17]	≥30 vs. 18.5–24.9	340	0.75 (0.56–1.00)	0.64 (0.47–0.86)
Naik GS, et al. [22]	25–35 vs. 18.5–24.9	139	0.43 (0.19–0.95)	0.26 (0.10–0.71)
Richtig G, et al. [23]	<25 vs. ≥25	76	1.03 (0.62–1.70)	1.81 (0.98–3.33)
Rutkowski P, et al. [24]	Continuous variable	446	1.00 (0.98–1.04)	1.02 (0.99–1.05)
Renal cell carcinoma				
Sanchez A, et al. [27]	≥30 vs. 18.5–24.9	203	NA	0.54 (0.31–0.95)
Lalani AA, et al. [28]	≥25 vs. <25	735	0.98 (0.80–1.20)	0.75 (0.57–0.97)
Santoni M, et al. [29]	≥25 vs. <25	675	NA	NA
Takemura K, et al. [32]	≥25 vs. <25	2281	0.66 (0.44–1.00)	0.77 (0.65–0.91)
Non-small cell lung cancer				
Cortellini A, et al. [33]	≥30 vs. 18.5–24.9	962	0.61 (0.45–0.82)	0.70 (0.49–0.99)
Kichenaddasse G, et al. [34]	≥30 vs. 18.5–24.9	1434	0.78 (0.62–0.96)	0.48 (0.34–0.66)
Ichihara E, et al. [35]	≥22 vs. <22	1429	0.79 0.64–0.98	0.73 (0.57–0.95)
Pan-cancer studies				
Yoo SK, et al. [37] (16 cancer types)	≥30 vs. 18.5–24.9	1840	0.77 (0.68–0.87)	0.81 (0.71–0.93)
Cortellini A, et al. [38]	≥25 vs. <25	976	0.51 (0.44–0.60)	0.33 (0.28–0.41)

### 2.4. Pan-Cancer Studies

Given that obesity was associated with improved outcomes with ICB in three cancer types with very different biologies, but all of which are quite responsive to ICB, a logical question is whether this relationship holds true across the other malignancies where ICB has not been approved. A large retrospective study including 1840 patients across 16 types of cancer examined the relationship between outcomes with ICB and pretreatment BMI as a categorical variable [37]. In that study, OS improved in a dose-dependent fashion whereby obese patients had better OS than overweight (HR 0.82, 95% CI 0.70–0.96) and normal BMI patients (HR 0.81, 95% CI 0.71–0.93, Table 1). Obese patients also had improved PFS with ICB when compared to normal BMI patients (HR 0.77, 95% CI 0.68–0.87). Overweight patients still had significantly longer OS (HR 0.81, 95% CI 0.71–0.93) and PFS (HR 0.86, 95% CI 0.76–0.97) than the normal BMI group. The authors also attempted to account for biological drivers of ICB response and stratified patients using tumor mutation burden (TMB), yet in both the high and low TMB groups, obesity remained associated with improved survival. Though no formal interaction testing was performed examining cancer subtype x BMI x outcome (as this would be underpowered for less represented cancers), the direction of the effect was the same across 14 of 16 malignancies and only in esophageal and head and neck cancer did non-obese patients have improved survival.

Another pan-cancer study suggests that it may be important to consider the class of obesity when considering outcomes with ICB. In a retrospective study of 976 patients with stage IV cancer treated with anti-PD-1/PD-L1 ICB, overweight and obese patients (BMI ≥ 25) had improved clinical outcomes compared to normal BMI patients [38]. Specifically, overweight and obese patients had improved time to treatment failure (TTF, HR 0.51, 95% CI 0.44–0.60) and OS (HR 0.33, 95% CI 0.28–0.41, Table 1). In multivariate analyses, TTF and PFS were not significantly associated with obesity (BMI ≥ 30), but they remained significantly associated with the overweight BMI cohort (BMI 25–30). This suggests there is a non-linear relationship between BMI and outcomes with ICB, and extremely elevated BMI may negatively impact clinical outcomes with ICB through its involvement in metabolic syndrome. Though this study was pan-cancer, 65% of patients had NSCLC, 19% melanoma, 14% RCC, and only 2.4% other.

Finally, a systematic review evaluated the influence of BMI on survival with ICB in 18 retrospective studies across cancer types including melanoma (n = 5), NSCLC (n = 5), and RCC (n = 3) [39]. The authors’ primary conclusion was that current evidence does not support a clearly positive association between clinical outcomes and BMI. As noted above, many of these studies have been positive, so variability in patient cohort and study design has made it difficult to discern a consistent impact of BMI on ICB outcomes. Future studies should focus on examining BMI and outcomes in the many other cancers in which ICB is now approved; however, the distribution of cancers reflected in the prior pan-cancer analyses reflects the overall “market share” for ICBs in specific cancer types, as well as when these drugs were approved and if they are used nearly universally in patients with a specific cancer type (e.g., melanoma) or only in defined subgroups (e.g., microsatellite unstable colorectal cancer).

### 2.5. Immune-Related Adverse Events

Obesity may impact both aspects of the therapeutic index for ICB as recent studies suggest that BMI may be associated with the incidence of immune-related adverse events (irAEs). A pooled, post hoc analysis of 3772 patients including eight tumor types (melanoma n = 832, RCC n = 1000, NSCLC n = 685, microsatellite instable colorectal cancer n = 193, hepatocellular carcinoma n = 214, Hodgkin lymphoma n = 342, head and neck cancer n = 236, urothelial carcinoma n = 270) treated with nivolumab with or without ipilimumab evaluated the incidence of irAEs by BMI [40]. For patients receiving nivolumab monotherapy (n = 2746), the incidence of any-grade irAEs was higher in those with obese BMI than normal or underweight BMI [odds ratio (OR) 1.71, 95% CI 1.38–2.11], but the incidence of more severe grade 3 or 4 irAEs did not differ by BMI. This finding was consistent across all subgroups, including tumor type. The incidence of irAEs was similar across BMI categories in patients who received nivolumab plus ipilimumab (n = 1026). Multiple retrospective studies have interrogated whether body composition parameters are associated with irAEs. In 84 patients with metastatic melanoma who received ipilimumab, low skeletal muscle quality was associated with grade 3 or 4 irAEs (OR 3.57, 95% CI 1.09–11.77) but sarcopenia was not [25]. In a study of 92 patients with primarily NSCLC treated with nivolumab, sarcopenia was associated with an increased but imprecise incidence of irAEs (OR 3.84, 95% CI 1.02–14.46) [41]. Notably, this study also evaluated nivolumab trough levels at day 14 and found that overweight and obese BMI was associated with higher trough levels (OR 5.94, 95% CI 1.25–28.29). Finally, a retrospective study of 100 patients with multiple tumor types treated with PD-1/PD-L1 ICB interrogated the relationship between skeletal muscle mass and quality with clinical outcomes [42]. In that study, SMI and muscle quality were not associated with irAEs, but the results were not conclusive due to the low incidence of irAEs. In sum, there is an association between increased BMI and increased irAEs that may be explained by concurrent sarcopenia or increased nivolumab concentrations. Further, the association between adipose compartments and irAEs remains unknown.

## 3. The Potential Influence of Obesity on Tumor and Immune Biology in the Setting of ICB

Given the growing interest in the obesity paradox in oncology and the high prevalence of obesity, recent preclinical and translational studies have explored how obesity may biologically impact response to systemic therapy. We and others have speculated that the biologic rationale for the obesity paradox may differ between tumor types given the differences in oncogenic signaling pathways and the tumor microenvironment, so we will initially discuss studies focused on melanoma followed by RCC. The first major translational study of the obesity paradox utilized specimens from multiple healthy species and diet-induced obese (DIO) mice with B16 melanoma to demonstrate that obesity was associated with increased PD-1 mediated T-cell dysfunction [43]. In similar pre-clinical models, the authors found that leptin signaling, a hormone produced by adipose tissue and positively correlated with obesity, in obese models increased PD-1 expression and promoted T-cell exhaustion (Figure 1). Finally, the authors identified similar markers of immune exhaustion in a subset of obese patients over the age of 60 with melanoma. These findings were provocative due the link between an adipokine, leptin, and PD-1 mediated T-cell dysfunction, which is targeted by ICB. However, this study also had significant limitations because the majority of the analyses did not use human tissue and the studies that were performed in human tissue were in selected populations. Our group subsequently published a study further investigating the obesity paradox using tissue from six cohorts and 765 patients with metastatic melanoma and patient-derived cell lines [44]. In summary, we found that metastatic melanoma tumors from overweight and obese patients had downregulation of oxidative phosphorylation (OXPHOS) and other metabolic pathways (Figure 1). The metabolic quiescence in tumors from OW/OB patients was observed across multiple lines of investigation, including bulk RNA sequencing and direct metabolic profiling using LC/MS and Seahorse bioenergetic analyses. Low OXPHOS and glycolysis may explain improved outcomes with ICB in obese patients because high OXPHOS has been shown to drive resistance to ICB via a hypoxic TME hostile to T-cell function [45,46]. Given the findings from Wang and colleagues, our study evaluated immune signatures in bulk RNA sequencing and by immunohistochemistry (IHC), but we were unable to replicate their findings of PD-1 mediated T-cell exhaustion in our tissue-based cohorts.

RCC is unique because histologic subtypes are metabolically distinct at the tumor level, and an obesity paradox has been described with ICB as well as VEGF-targeted therapies [47]. To date, translational studies investigating the biologic rationale for the obesity paradox in RCC have focused on outcomes with VEGF-targeted therapies, yet these studies provided novel insights into how the host metabolism may influence the RCC tumor and TME. In an analysis of 126 patients with stages I-III clear cell RCC, obese patients had decreased expression of FASN, an enzyme involved in fatty acid anabolism, and increased expression of pathways involved in fatty acid metabolism and oxidation [48]. This work led to an analysis of 61 patients with metastatic clear cell RCC that confirmed FASN gene expression was inversely correlated with BMI, and in a cohort of 146 patients treated with targeted therapy, higher FASN expression by IHC was associated with inferior overall survival [49]. A subsequent translational study further evaluated the biologic rationale for the obesity paradox in clear cell RCC in 375 patients across multiple cohorts [27]. In a gene set enrichment analysis (GSEA) of bulk RNA sequencing, obese tumors had upregulation of angiogenesis, wound healing, and metabolic pathways including fatty acid metabolism. Most notably, the authors evaluated peri-tumoral adipose tissue using RNA sequencing and found that obese patients had higher immune infiltration and hypoxia in these adipose reservoirs than their normal BMI counterparts, which may account for the obesity paradox observed with ICB in metastatic ccRCC.

A significant challenge with modeling the impact of obesity on ICB outcomes is the translatability of obesity in model organisms to humans [50]. In mice, the obese phenotype is generated either by feeding standard mice a very high fat diet (i.e., diet-induced obesity) *or* utilizing mice with genetic mutations that lead to obesity (e.g., leptin or its receptor) [51]. Murine tumor cells are then injected into the mice and the mice are treated with the agent of interest. However, neither of these models recapitulate the complex biology and etiology of human obesity.

## 4. Analytical Considerations for the Obesity Paradox

Despite growing evidence in support of an obesity paradox with ICB in select malignancies, the causal relationship between obesity and clinical outcomes with ICB is questioned by some because many clinical studies overlook key analytical considerations. A limitation of some studies evaluating the obesity paradox with ICB is the lack of a clearly defined hypothesis or the absence of clearly defined causal relationships among the variables studied. Causal diagrams such as directed acyclic graphs (DAGs) are tools that enable clinical researchers to visually define and communicate the modeled relationship between exposure (BMI), outcomes (survival), and potential mediators, colliders, and confounders [52,53]. Without defining these relationships, analyses may not condition for potential confounding variables or inappropriately condition for collider variables, which both increase bias and may lead to misassignment of causality [47,54]. Studies evaluating obesity as an exposure are at risk for two biases that are best understood through DAGs, the collider stratification bias and reverse causality. In scenarios where a patient having a malignancy is a collider variable, restricting the analysis to only patients with that malignancy induces a collider stratification bias. Figure 2A shows how studies evaluating BMI in patients with a select malignancy such as NSCLC can open a noncausal backdoor pathway between confounding variables, such as smoking, and survival that results in BMI falsely appearing protective [55,56,57]. NSCLC acts as a collider of BMI and smoking. Therefore, selecting only patients with NSCLC in an analysis induces the risk for collider stratification bias. This bias can be accounted for by adjusting for smoking which will block the noncausal backdoor pathway. An additional methodologic consideration is that studies evaluating BMI or weight at a single time point, such as at ICB initiation, are at risk for reverse causality. If weight change over time pre- and post-diagnosis is not accounted for then the confounding effect of cancer-related weight change due to cachexia can spuriously impact survival estimates (Figure 2B) [55,58].

Most studies evaluating the obesity paradox utilize BMI to define obesity, which has limitations that impact the causal inferences we can draw from these studies. BMI is an imperfect surrogate to measure body composition, but it has been widely used due to the ease of obtaining it from routine clinical practice. Additionally, BMI is often reported as a categorical variable in clinical studies, instead of a continuous variable, which results in a loss of information, and it also leads to an unstated assumption that a BMI of 30.5 produces the same host metabolic milieu as a BMI of 50. Body composition, which includes subcutaneous adiposity, visceral adiposity, and skeletal muscle mass and quality, can be measured directly using imaging modalities, such as computed tomography (CT) or dual-energy X-ray absorptiometry (DEXA) [59,60]. Studies that report the relationship between direct measurements of body composition and clinical outcomes with ICB facilitate more discreet biological hypotheses and improved reproducibility. For example, BMI does not provide information on whether the obesity paradox is driven by the interaction between visceral or subcutaneous adipose tissue and the tumor or by muscle mass. Thus, it remains possible that adiposity is not the driver of the obesity paradox, but instead, it is the presence of sarcopenia that leads to poor outcomes.

## 5. The Future of the Obesity Paradox with ICB

While further research is needed to understand the mechanism of how host obesity influences a tumor, it stands to reason that multiple host variables, including obesity, may interact with the tumor cell and the TME [61]. Other cardiometabolic diseases, such as diabetes mellitus, hypertension, or coronary artery disease, can alter systemic metabolism via excess nutrients, growth factors, and altered peptides that may impact cancer biology. For example, pre-clinical studies suggest that metformin, a drug for diabetes, can metabolically reprogram the TME and augment the efficacy of ICB, and early clinical studies suggest a potential for improved survival outcomes in patients with melanoma who receive metformin [62,63,64,65]. There is also a growing literature suggesting that sex hormones can influence response to systemic therapy, including ICB. Androgen deprivation therapy (ADT) has been shown to improve thymic output in patients, and in pre-clinical models, ADT plus enzalutamide sensitizes tumors to ICB by enhancing CD8 T-cell function [16,66]. In pre-clinical models of hormone receptor-positive breast cancer, chemoprevention with tamoxifen, an anti-estrogen, is dependent upon functional T cells [67]. However, a phase III clinical trial evaluating enzalutamide plus pembrolizumab was discontinued because radiographic PFS and OS were not improved, which highlights the challenges of translational research for host variables, as previously discussed. Advancing age is an established risk factor for the incidence of cancer, and age is associated with profound biological changes, such as senescence and chronic inflammation [68,69]. Studies have suggested that increasing age is associated with improved or comparable response to ICB, but not inferior clinical outcomes [70,71]. Finally, it is difficult to dissect the determinants of obesity (diet, exercise, and genetic factors), the microbiota which are responsive to these factors, and the effects of obesity on ICB. Advances in spatial technologies will accelerate pre-clinical and translational investigation into the complex interaction between the three dimensions that influence ICB response, tumor intrinsic, TME, and host ecological variables, but moving forward, disciplined advances in clinical research methodology are needed to begin elucidating the role of host variables in response to ICB and other systemic therapies.

## 6. Conclusions

Obesity, defined by BMI, is associated with improved survival when patients with multiple advanced cancers are treated with ICB, including melanoma, RCC, and NSCLC. Early investigations into the underlying biology suggest that obesity may alter tumor metabolism, TME immune cell composition, and angiogenesis, but a causal mechanism has not been established for the obesity paradox. As interest in the host tumor interactome grows, it is critical that clinical and translational studies account for non-causal pathways identified by DAGs and are mitigated by appropriate statistical conditioning.

## Figures and Tables

**Figure 1 cells-12-02551-f001:**
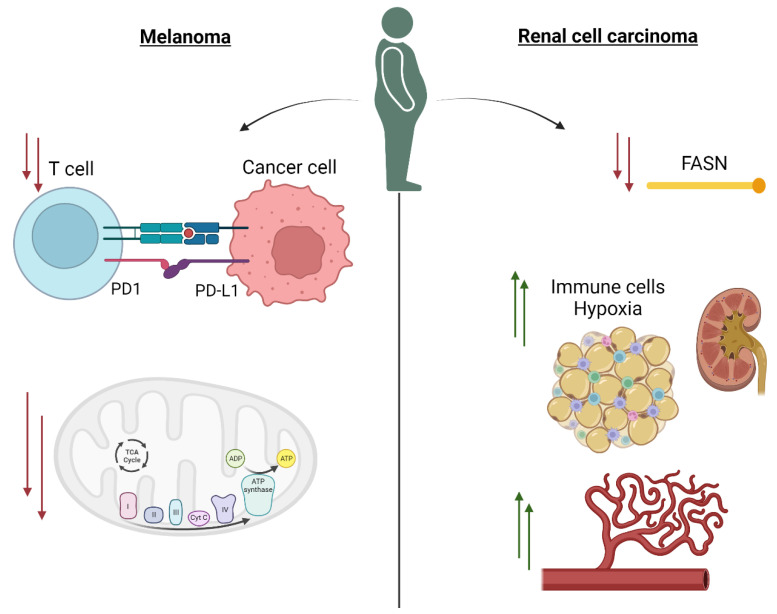
Title: Potential biological mechanisms underlying the obesity paradox with immune checkpoint therapy (ICB). Legend: In melanoma, pre-clinical models suggest obesity is associated with PD-1 mediated T-cell dysfunction that may be driven, in part, by leptin, an adipokine. In human tissue-based studies, obesity or overweight body mass index (BMI) is associated with decreases in oxidative phosphorylation and associated with a metabolically quiescent state. In renal cell carcinoma, obesity is associated with decreased expression of FASN, increased immune infiltration and hypoxia in peritumoral adipose tissue, and increased angiogenesis.

**Figure 2 cells-12-02551-f002:**
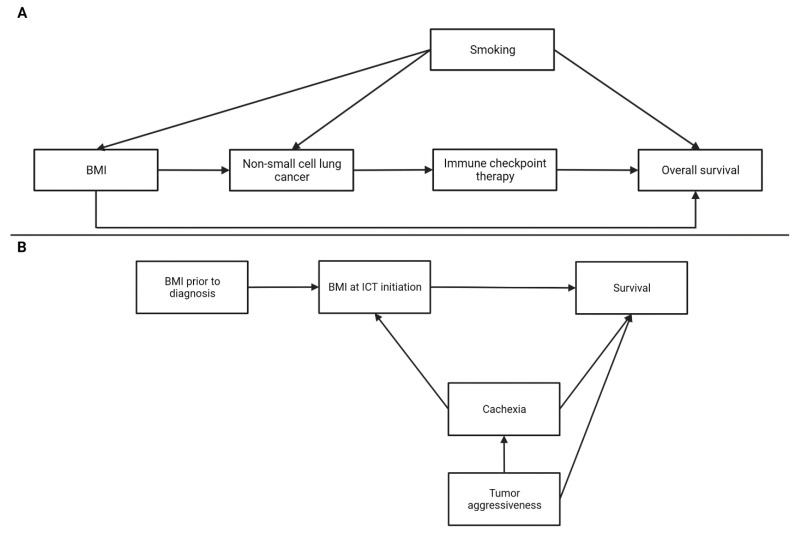
Title: Directed acyclic graphs (DAGs) depicting analytical considerations for studying the obesity paradox with immune checkpoint therapy (ICB). Legend: (**A**) The potential for collider stratification bias when studying the obesity paradox. This figure shows how cancers such as non-small cell lung cancer (NSCLC) may act as colliders when BMI and smoking increase cancer risk. In these cases, if we restrict our analyses only to patients with the cancer, such as NSCLC, then a non-causal pathway is opened whereby BMI spuriously influences overall survival via smoking. However, if smoking is adjusted for then the non-causal pathway will be closed thus allowing the more reliable estimation of the effect of BMI on survival. (**B**) Potential for reverse causality when studying the obesity paradox by not adjusting for the presence of cancer-related cachexia that can impact the exposure (BMI at ICB initiation) and outcome (survival). Confounding variables, such as cachexia, should be adjusted for in analyses. If the presence of cachexia is not clear, then BMI prior to diagnosis is a useful additional data point not impacted by cachexia.

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
