# Peer review of "The Influence of Obesity on Outcomes with Immune Checkpoint Blockade: Clinical Evidence and Potential Biological Mechanisms"

_cells, 2023, doi:10.3390/cells12212551_

Round 1

Reviewer 1 Report

Comments and Suggestions for Authors

1. Abstract: Change "biologic" to "biological". 

2. In the closing paragraph of the Introduction, the authors must define their search strategy and inclusion and exclusion criteria for relevant studies.

3. The authors should provide a rationale why the focus was on three types of cancer described in detail.

4. Among the less discussed toxicities of ICB is aplastic anemia (PMID: 36778017, 36459923, 36852421). The authors are encouraged to cover that aspect as well.

5. Figure 1 and 2 cannot be named "Legend". Amend.

Reviewer 2 Report

Comments and Suggestions for Authors

The paper presents the effects of obesity on the effects of immunotherapy in cancer patients.

The presentation of data is not summarized in tables, which makes it difficult to follow.

The abstract is not informative and does not present data that supports the 'obesity paradox'.

In my opinion, a better effect of checkpoint inhibitors in patients with obesity may reflect a lack of immunological exhaustion. Perhaps malnutrition determines the poor response.

Therefore, it seems better to avoid the term obesity paradox and change the title to - obesity as a potential predictor of therapy with checkpoint inhibitors in palliative systemic therapy.

The chapter 'Potential biological mechanisms underlying the obesity paradox' is highly speculative. Perhaps it would be better to rewrite it to 'Potential mechanisms explaining the predictive role of obesity during checkpoint inhibitors therapy.

Round 2

Reviewer 2 Report

Comments and Suggestions for Authors

The paper has been improved. It is true that the mechanism behind the association is unknown.  

No further comments.